DATA RELEASE

# A new and improved genome sequence of *Cannabis sativa*

Shivraj Braich[1,2], Rebecca C. Baillie[1], German C. Spangenberg[1,2] and Noel O. I. Cogan[1,2,*]

1 Agriculture Victoria, AgriBio, Centre for AgriBioscience, Bundoora, Victoria 3083, Australia
2 School of Applied Systems Biology, La Trobe University, Bundoora, Victoria 3086, Australia

## ABSTRACT

Cannabis is a diploid species ($2n$ = 20), the estimated haploid genome sizes of the female and male plants using flow cytometry are 818 and 843 Mb respectively. Although the genome of Cannabis has been sequenced (from hemp, wild and high-THC strains), all assemblies have significant gaps. In addition, there are inconsistencies in the chromosome numbering which limits their use. A new comprehensive draft genome sequence assembly (~900 Mb) has been generated from the medicinal cannabis strain Cannbio-2, that produces a balanced ratio of cannabidiol and delta-9-tetrahydrocannabinol using long-read sequencing. The assembly was subsequently analysed for completeness by ordering the contigs into chromosome-scale pseudomolecules using a reference genome assembly approach, annotated and compared to other existing reference genome assemblies. The Cannbio-2 genome sequence assembly was found to be the most complete genome sequence available based on nucleotides assembled and BUSCO evaluation in *Cannabis sativa* with a comprehensive genome annotation. The new draft genome sequence is an advancement in Cannabis genomics permitting pan-genome analysis, genomic selection as well as genome editing.

**Subjects** Genetics and Genomics, Plant Genetics, Botany

# MAIN CONTENT

## Context

The legalisation of medicinal cannabis has spread across the globe leading to increased benefits for a range of conditions. *Cannabis sativa* (NCBI:txid3483) is an erect, annual, wind-pollinated herb, that is typically dioecious although monoecious forms can exist. The plant is diploid ($2n$ = 20) with gender driven by a pair of sex chromosomes (X and Y) along with the nine autosomes [1, 2]. The diploid genome sizes of the female and male plants using flow cytometry are 1636 ± 7.2 and 1683 ± 13.9 Mbp, respectively [3, 4]. Cannabis plants are best known for cannabinoid biosynthesis, most prominent of these include delta-9-tetrahydrocannabinol ($\Delta^9$-THC, or simply THC) and cannabidiol (CBD). Preparations from medicinal cannabis extract have various pharmacological effects (depending on the cannabinoid composition) for example, CBD has effects as a muscle relaxant, anticonvulsant, neuroprotective, antioxidant, anxiolytic and also has antipsychotic activity; while THC's effects can be utilised as a psychopharmaceutical, as well as an analgesia, appetite stimulation, antiemesis and also for muscle relaxation [5]. Besides CBD and THC, other cannabinoids such as cannabichromene (CBC) [6], cannabigerol (CBG) [7] and delta-9-tetrahydrocannabivarin (THCV) [8] have also been recognised to have pharmacological effects. Moreover, secondary metabolites from cannabis plant tissues,

**Submitted:** 13 October 2020

\* Corresponding author. E-mail: noel.cogan@agriculture.vic.gov.au

Preprint submitted at https://doi.org/10.1101/2020.12.13.422592

such as flavonoids and terpenes are also known to contribute to psychoactive or therapeutic effects [9]. The biosynthesis of cannabinoids and terpenes with medicinal properties is currently only partly understood and additional genetic and genomic studies will further illuminate the different production mechanisms that the various plant genotypes deliver.

An initial draft genome sequence of cannabis was published in 2011 that generated 534 Mbp of assembled nucleotides available from the drug-type variety, Purple Kush (PK) [10]. Following the generation of an initial draft genome sequence, several chromosome-scale whole genome sequence assemblies were made available in 2018 using long-read sequencing technology from the strains; PK (high THC producing female plant, GenBank-GCA_000230575.5), Finola (hemp, male plant, GenBank-GCA_003417725.2) and CBDRx (high CBD producing plant, genome sequence assembly named cs10 within GenBank-GCA_900626175.2) and recently in 2020 from the strain, JL (wild-type, female plant, GenBank-GCA_013030365.1) with assembled sequence size of 639 Mb, 784 Mb, 714 Mb and 797 Mb, respectively (without Ns) [11–13]. Despite the use of long-read sequencing technology, the published assemblies have significant gaps and inconsistent nomenclature of chromosomes numbering and orientation. The availability of a comprehensive genome sequence from a medicinal strain will add clarity relating to gene characterisation and functional analysis as well as valuable diversity for a pan-genome analysis.

The current study reports the development of an improved comprehensive draft genome sequence for *Cannabis sativa* that integrates the dataset generated from a female genotype which produces a balanced CBD:THC cannabinoid ratio, Cannbio-2 (Cb-2, Figure 1, [14]). The study also provides the genome annotation using the published extensive transcriptome dataset [15] as evidence and evaluation of the generated genome sequence and compares the sequence dataset to available whole genome sequence assemblies.

## METHODS

### Plant materials and DNA isolation

All plants were maintained under artificial conditions in controlled environment facilities and all the work undertaken was performed under Medicinal Cannabis Research Licence (RL011/18) and Permit (RL01118P6) issued by the Department of Health (DoH), Office of Drug Control (ODC) Australia. A variety of seeds were imported from a legal source in Canada and were screened with DNA markers and using comprehensive chemical analysis [14]. Cannbio-2 was identified as a female plant and selected as an optimal strain that produces a balanced CBD:THC cannabinoid ratio [14]. Fresh leaves were sampled from the female cannabis plant, Cannbio-2, and the harvested tissue was stored at −80 ˚C until required. Genomic DNA was isolated with the DNeasy® Plant 96 Kit (QIAGEN, Hilden, Germany) following manufacturer's instructions. Isolated high molecular weight DNA was quantified by fluorometry (Qubit, Thermo Fisher Scientific, Waltham, USA) and assessed for quality using a 1% (w/v) pulse-field gel electrophoresis and with genomic ScreenTape on the TapeStation 2200 platform (Agilent Technologies, Santa Clara, CA, USA).

### Pacific Biosciences sequencing and genome assembly

Single Molecule Real Time (SMRT) bell libraries were prepared from the extracted DNA using the SMRTbell™ Template Prep Kit 1.0-SPv3 according to the protocol "20 kb Template Preparation Using BluePippin Size-Selection System" as recommended by the manufacturer (Pacific Biosciences) with the exception that the initial DNA was not sheared. Incompletely

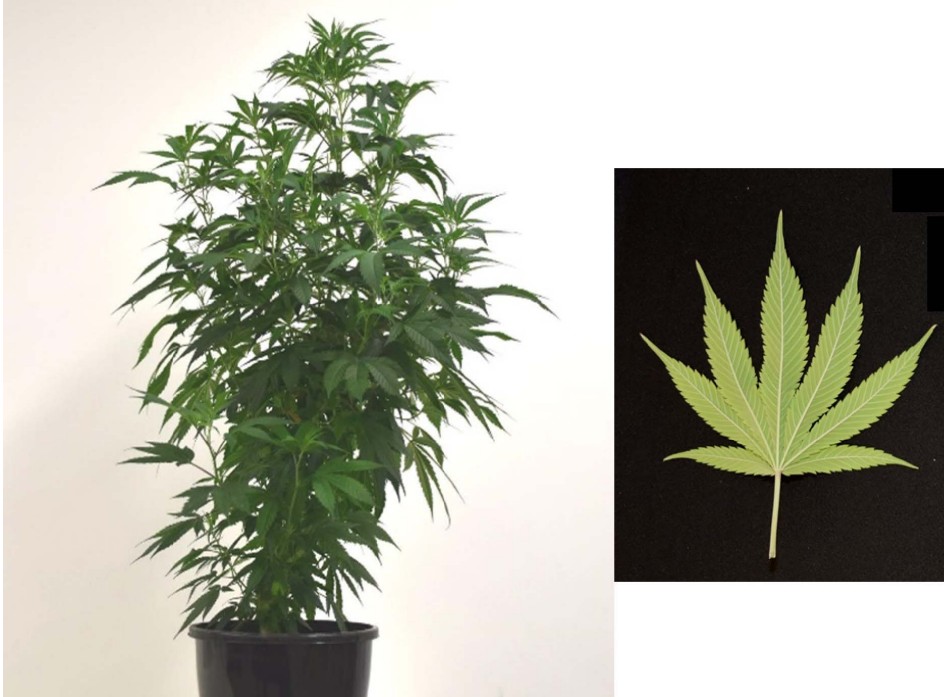

**Figure 1.** Example of Cannbio-2 plant with its leaf characteristics.

formed or non-SMRTbell DNA was removed by exonuclease treatment. The SMRTbell templates were size-selected using the BluePippin system (Sage Sciences) on a 0.75% (w/v) agarose gel cassette aiming to remove library insert sizes smaller than 15 kb. Size-selected libraries were further cleaned using the AMPure PB beads (Pacific Biosciences). The SMRTbell templates were quantified by a high-sensitivity fluorometric assay (Qubit, Thermo Fisher Scientific, Waltham, USA) and quality assessed using Genomic DNA ScreenTape on the TapeStation 2200 platform (Agilent Technologies, Santa Clara, CA, USA). The generated SMRT bell templates were sequenced on the PacBio Sequel instrument (PacBio Sequel System, RRID:SCR_017989) with the Sequel™ SMRT® cells 1M v2 Tray as per the manufacturer's instructions. The raw PacBio reads were error-corrected and assembled using the SMRT Link's Hierarchical Genome Assembly Process (HGAP4) *de novo* assembly application (v5.0.0) with default parameters to generate the *de novo* assembly. RaGOO [16] (v1.1) that uses minimap2 (v2.10, RRID:SCR_018550) [17] was used to reference align, to order and orientate the draft genome assembly contigs of Cannbio-2 to chromosome scale pseudomolecules using reference genomes of cs10, PK and JL. Default parameters with the exception of the "-b" option, to break chimeric contigs and "-g 100" to use gap size of 100 N's for padding in pseudomolecules was used.

## Comparison of genome assemblies

Available whole genome assemblies of cs10, PK, Finola and JL were compared to the generated genome assembly in the current study. For the comparisons, whole-genome sequence alignments were created using minimap2 [17] (v2.10) with the parameter "-x asm5–cs" to generate pairwise alignment format (PAF) file using the Cannbio-2 genome



sequence assembly as the reference and published genome sequence assemblies as query. The alignments were converted to dot plot using dotPlotly v1.0 [18] in R.

## Genome annotation

The genome annotation was performed following the GenSAS [19] v6 pipeline on the draft assembly contigs ordered into pseudomolecules. Repeat regions in the genome assembly were masked using RepeatMasker v4.0.7 (RRID:SCR_012954) [20] (with "*Arabidopsis thaliana*, *Oryza sativa* and other dicots" repeat libraries) and *de novo* repeat finding tool RepeatModeler v1.0.11 (RRID:SCR_015027) [21] to create a soft-masked consensus sequence. Transcript alignments were generated using BLASTN (v2.7.1, RRID:SCR_001598), BLAT (v35, RRID:SCR_011919) and PASA (v2.3.3, RRID:SCR_014656) using the Cannbio transcriptome assembly [15] as the database (BioProject: PRJNA560453, BioSample: SAMN13503240-SAMN13503310, SRA: SRR10600874-SRR10600944). Initial *ab initio* gene predictions were made using Augustus (v3.3.1, RRID:SCR_008417) [22] with species "*Arabidopsis thaliana*". EVidenceModeler (EVM, v06/25/2012, RRID:SCR_014659) [23] was used to create the consensus gene set by combining gene predictions from Augustus (weight score-1) and results from transcripts alignments (weight score-10). The consensus gene set was further refined using PASA to create the final gene set which was used for functional annotation. Functional analysis of the final gene set was primarily conducted using DIAMOND (v0.9.22, RRID:SCR_016071) [24] analysis to SwissProt database. Putative THCAS/CBDAS genes were identified based on the annotation and plotted across the genome using karyoplyteR (v1.10.0) [25] in R. Other tools were also utilised for the functional analysis including InterProScan (v5.25-68.0, RRID:SCR_005829) [26] and Pfam (v1.6, RRID:SCR_004726) [27]. The results from functional analysis were merged in creating an annotated genome submission in a GFF3 format.

## RESULTS AND DISCUSSION

### Generation of genome sequence assembly

Cannbio-2 was sequenced to 86× genome coverage by generating 70.09 Gbp of sequence data. The draft sequence assembly generated by HGAP4 resulted in 8477 contigs assembled in 913.5 Mb with maximum contig length of 1705,170 bp and N50 of 187,352 bp (Table 1). The draft genome sequence assembly of Cannbio-2 was comprehensively analysed through a reference guided assembly approach using the published genome sequence assemblies of PK, cs10 and JL as references to guide the chromosome scale sequence assembly process, resulted in genome assembly sizes (with Ns) of 756.33 Mb, 904.08 Mb and 891.96 Mb respectively (Table 2). Cannbio-2 genome sequence assembly guided using cs10 genome sequence assembly was found to be the largest based on nucleotides assembled and was used for subsequent analysis to compare the draft genome to the other available references. Furthermore, cs10-guided assembly was also chosen for further analysis due to its chromosome nomenclature (which uses the linkage groups nomenclature from a previous study [28]). The statistical analysis of the new genome assembly generated from the current study and previously published chromosome-scale genome assemblies are summarised in Table 1. The analysis revealed that the generated genome sequence was found to be the most complete with assembly size of 903 Mb when compared to the whole genome assemblies of cs10 (714 Mb), Finola (784 Mb), PK (639 Mb) and JL (797 Mb). The size of the generated genome assembly was found to be larger than the estimated *C. sativa* (Hemp)



**Table 1.** Statistics of Cannbio-2 genome assembly from the current study as compared to published whole genome sequence assemblies.

| Data type | Cb-2$^d$ | Cb-2$^r$ | cs10 | JL | Finola | PK |
|---|---|---|---|---|---|---|
| Number of contigs/scaffolds | 8477 | 10 | 10 | 10 | 10 | 10 |
| Assembly size with Ns (Mb) | 914 | 904 | 854 | 798 | 785 | 640 |
| Assembly size without Ns (Mb) | 914 | 903 | 714 | 797 | 784 | 639 |
| Largest contig/scaffold (Mb) | 1.7 | 106 | 105 | 93 | 101 | 79 |
| N50 (Mb) | 0.2 | 91 | 92 | 83 | 87 | 72 |
| N90 (Mb) | 0.05 | 72 | 65 | 69 | 50 | 51 |

$^d$ Draft Cb-2 genome assembly. $^r$ RaGOO assigned Cb-2 genome assembly using cs10 as the reference.

**Table 2.** Number of bases per chromosome of Cannbio-2 genome assembled guided by PK, cs10 and JL genome assembly as the reference.

| Sequence | PK-guided assembly | cs10-guided assembly | JL-guided assembly |
|---|---|---|---|
| Cs_Cb2_01 | 91,352,534 | 86,898,403 | 104,860,357 |
| Cs_Cb2_02 | 84,314,258 | 105,265,154 | 105,786,500 |
| Cs_Cb2_03 | 89,716,256 | 87,707,768 | 91,501,419 |
| Cs_Cb2_04 | 85,532,416 | 100,932,893 | 92,102,208 |
| Cs_Cb2_05 | 84,300,950 | 91,493,340 | 95,601,317 |
| Cs_Cb2_06 | 72,493,431 | 97,797,982 | 89,863,944 |
| Cs_Cb2_07 | 75,583,091 | 85,051,101 | 92,903,079 |
| Cs_Cb2_08 | 72,000,744 | 71,555,044 | 79,110,046 |
| Cs_Cb2_09 | 62,999,750 | 71,141,854 | 76,841,943 |
| Cs_Cb2_10 | 38,036,213 | 106,236,836 | 63,393,006 |
| Total assembled size (Mb) | 756,329,643 | 904,080,375 | 891,963,819 |

genome size using flow-cytometry (818 Mb) [4]. The differences in the genome size could possibly reflect bias introduced due to the use of a different accession to orient and order the contigs to pseudomolecules or potential haplotype duplication or the genome variations (such as insertions, inversions, tandem repeats to name a few) between the hemp and medicinal cannabis strain.

## Comparison of genome assemblies

The generated genome assembly was found to be consistent in terms of chromosome nomenclature with few structural differences based on the alignment results when compared to the cs10 genome assembly (Figure 2). Despite the larger size of the generated genome assembly, large regions of duplication were not apparent when alignments were visualised as represented in Figure 2, highlighting the contiguity of the generated assembly. Comparisons were also made between Finola, PK and JL to the generated genome based on the alignment results (Figure 3, 4 and 5; Alignment files in GigaDB [29]). The comparisons of the genome sequences revealed large pericentromeric differences and chromosome inversions between the Cannbio-2 genome sequence and the genome sequences of Finola, PK and JL. Moreover, comparisons of JL, cs10, PK and Finola genome sequences revealed inconsistencies between these genome sequences in terms of orientation and numbering of chromosomes (Alignment files in GigaDB [29]).

## Genome annotation

The total predicted features from the repeat-masked consensus sequence were found to be 3419,223. Initial *ab initio* gene predictions that were made using Augustus resulted in prediction of 40,633 genes. The consensus gene set, derived by EVidenceModeler, generated



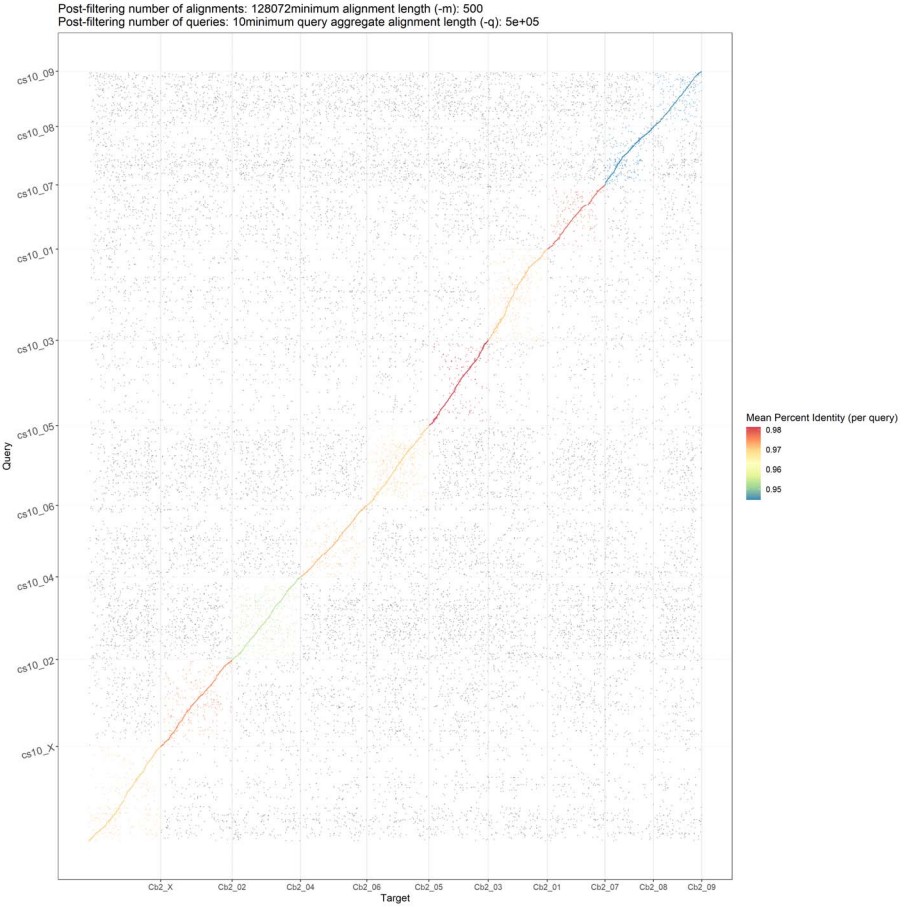

Post-filtering number of alignments: 128072minimum alignment length (-m): 500
Post-filtering number of queries: 10minimum query aggregate alignment length (-q): 5e+05

**Figure 2.** Dot plot showing alignments of Cannbio-2 sequence assembly to the whole genome sequence assembly of cs10.

a prediction of 36,758 genes which was further refined using PASA. The total predicted features from the final gene set following PASA refinement were 109,686 with 36,632 genes, 37,107 mRNA and 35,947 proteins. The predicted features per chromosome are as summarised in Table 3. Figure 6 represents the karyoplot of the density of masked repeats and genes across the 10 chromosomes of the Cannbio-2 annotated genome. Functional analysis of the final gene set based on DIAMOND analysis to SwissProt database, resulted in the identification of 16 putative THCAS/CBDAS genes across the Cannbio-2 genome sequence with 12 of these genes coded by chromosome 7 (Figure 6).

## DATA VALIDATION AND QUALITY CONTROL

Genomic DNA was extracted from fresh leaves of the Cannbio-2 plant using the DNeasy 96 Plant Kit (QIAGEN, Hilden, Germany), according to the manufacturer's instructions. Whole genome of Cannbio-2 was re-sequenced using an enzymatic MspJI (NEB, MA, United States) shearing method [30] as described previously [31]. The library was assessed using a D1000 ScreenTape on the TapeStation 2200 (Agilent, Santa Clara, CA, USA) and was subjected to paired-end sequencing on a HiSeq 3000 instrument (Illumina Inc., San Diego, CA, USA). The initial generated fastq sequences were quality trimmed using a custom perl script

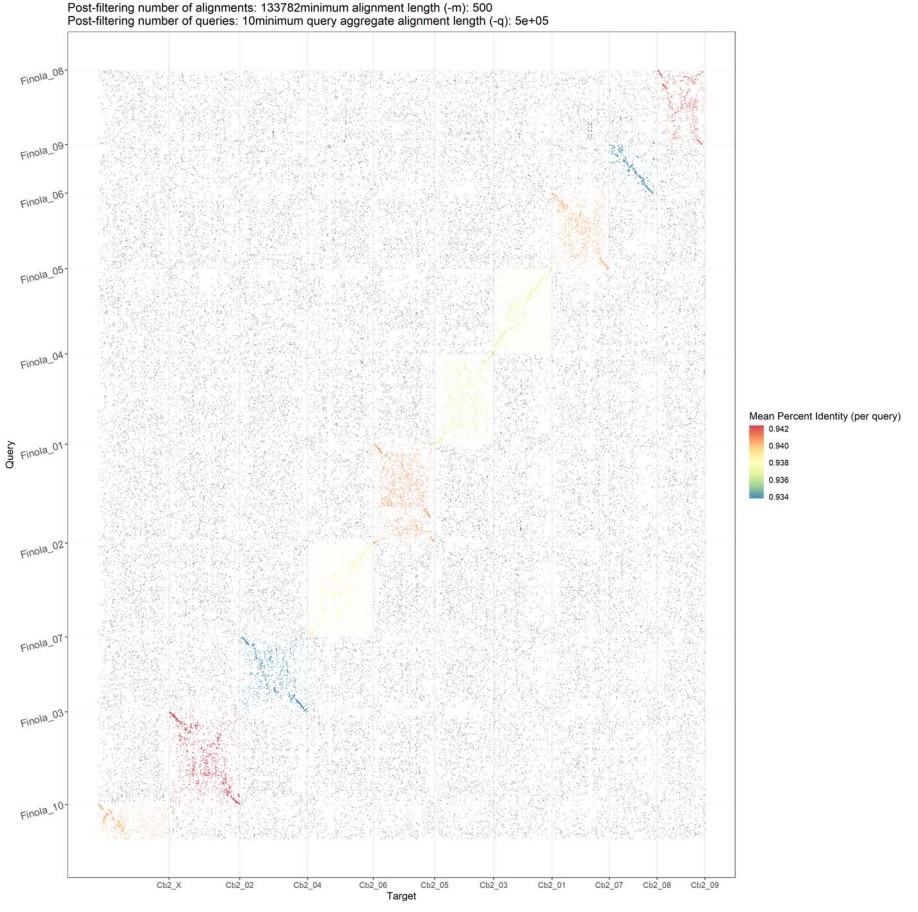

**Figure 3.** Dot plot showing alignments of Cannbio-2 sequence assembly to the whole genome sequence assembly of Finola.

**Table 3.** Number of predicted features following repeat masking and following genome sequence annotation (protein, mRNA and gene) per chromosome of the Cannbio-2 genome sequence assembly.

| Sequence name | Predicted features—Repeats | Predicted features—Annotation |
|---|---|---|
| Cs_Cb2_01 | 358,706 | 15,722 |
| Cs_Cb2_02 | 385,162 | 11,967 |
| Cs_Cb2_03 | 310,124 | 9,171 |
| Cs_Cb2_04 | 389,512 | 11,461 |
| Cs_Cb2_05 | 335,407 | 9,121 |
| Cs_Cb2_06 | 345,018 | 9,335 |
| Cs_Cb2_07 | 308,329 | 9,693 |
| Cs_Cb2_08 | 286,461 | 10,872 |
| Cs_Cb2_09 | 283,015 | 9,558 |
| Cs_Cb2_10/X | 417,489 | 12,786 |
| Total | 3419,223 | 109,686 |

(available in GigaDB [29]) and adaptor trimmed by cutadapt (v2.6, RRID:SCR_011841) [32]. The trimmed sequence reads were aligned to the generated sequence assembly of the Cannbio-2 strain using the BWA-MEM software package [33] (v0.7.17, RRID:SCR_010910) with default parameters, to evaluate the genome assembly. The alignment results of the

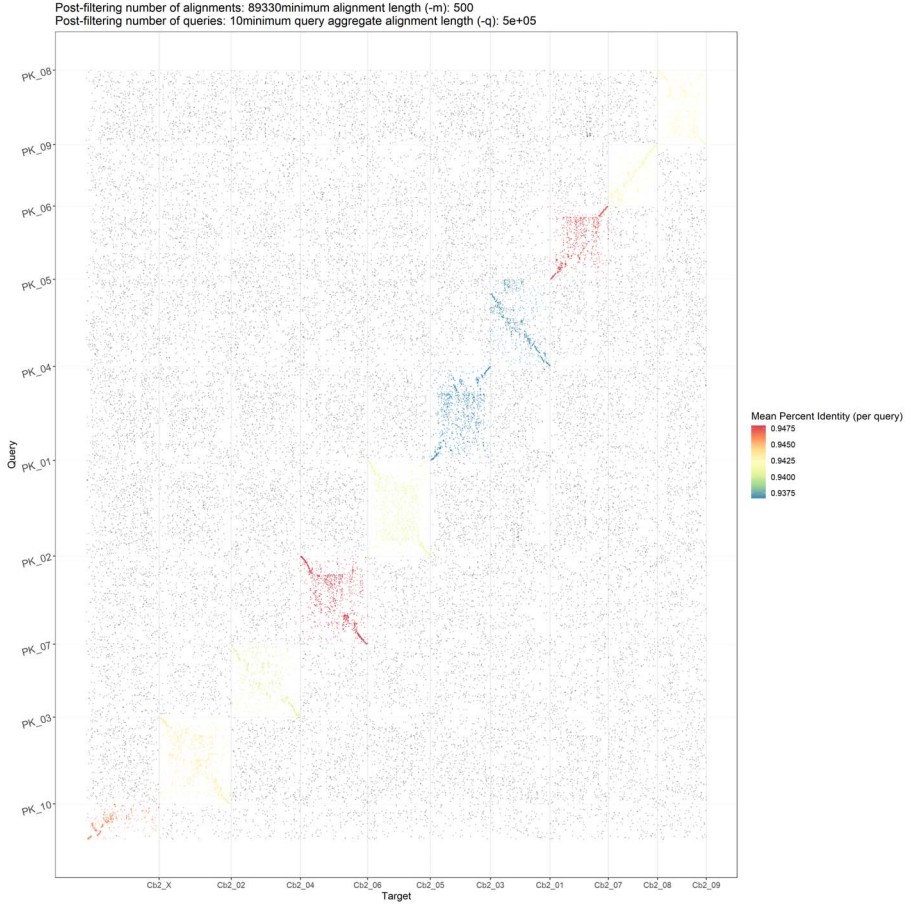

**Figure 4.** Dot plot showing alignments of Cannbio-2 sequence assembly to the whole genome sequence assembly of PK.

sequence reads to the generated genome assembly indicated that out of a total of 178.72 million QC-passed reads, 99.65% sequence reads were found to be mapped with 86.78% of sequence reads being properly paired, suggesting that the generated genome assembly contained comprehensive genomic information.

Benchmarking Universal Single-Copy Orthologs (BUSCO, v4.0.6, RRID:SCR_015008) [34] approach was used with the eudicotyledons_odb10 dataset in genome mode for all the genome assemblies to assess the completeness of the conserved proteins in the published and current genome sequence assemblies. Only pseudomolecules were used in the BUSCO analysis across all the genomes. The Cannbio-2 genome sequence captured 93% of genes as predicted by BUSCO evaluation which was found to be higher than all other published genome assemblies of (cs10-90.3%; Finola-82.6%; JL-86.5%; PK-78.2%; Figure 7). The results from the BUSCO analysis confirms the completeness of the Cannbio-2 genome sequence assembly. Furthermore, a detailed BLASTN analysis (v2.9.0) was performed to search for inadvertent chloroplast (KR184827.1, 153,848 bp and NC_027223.1, 153,854 bp) and mitochondrial (KR059940.1, 414,545 bp) genomes to check for the integration of organellar genomes in the generated assembly. The similarity results of Cannbio-2 genome to the

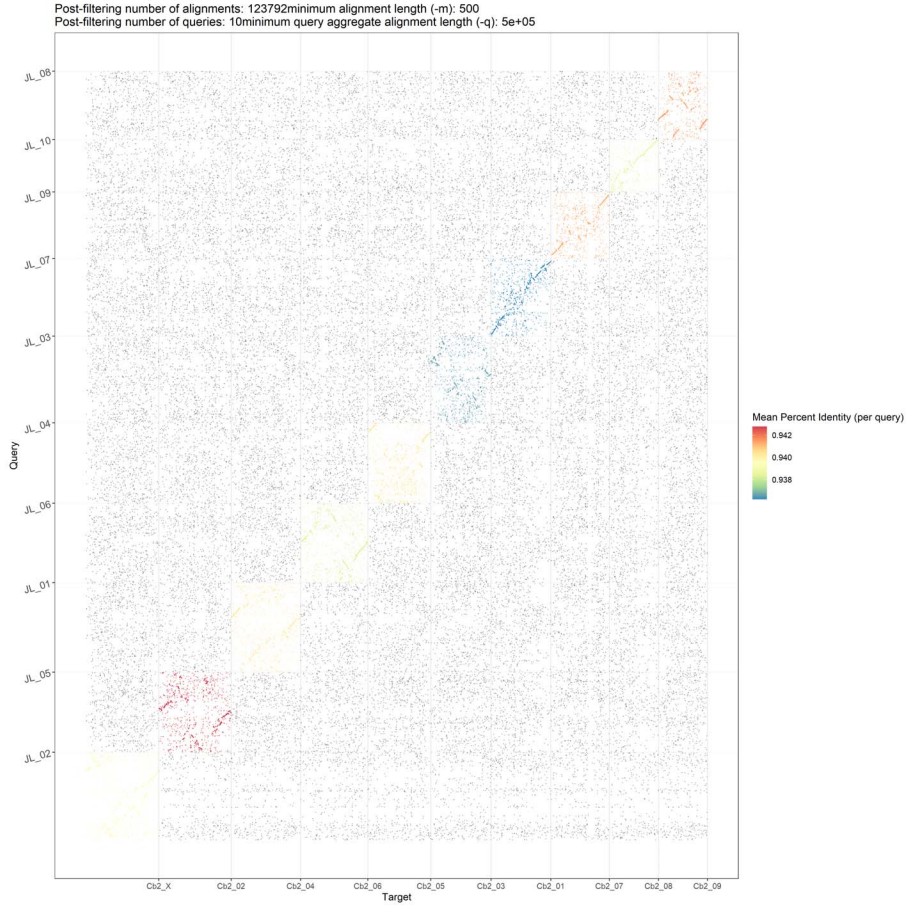

**Figure 5.** Dot plot showing alignments of Cannbio-2 sequence assembly to the whole genome sequence assembly of JL.

organellar genomes showed incorporation of small fragments with a maximum length of 30 kb for chloroplast genome sequence assembly and 12 kb for mitochondrial genome sequence assembly (BLASTN results in GigaDB [29]). The similarity results suggest no significant integration of these inadvertent genome sequences into the Cannbio-2 genome sequence assembly.

## CONCLUSION AND FUTURE PERSPECTIVE

The results suggest that the Cannbio-2 draft genome is the most comprehensive genome sequence of cannabis published to date. The development of a contiguous cannabis genome sequence will provide novel insights into the identification of genome-wide sequence variants. The research from the current study will also enable genomic selection, genome editing and pan-genome sequence analysis in medicinal cannabis.

## DISCLAIMER

The genome sequence data generated in this study was not assessed for the presence of potential haplotype duplication and genome heterozygosity.

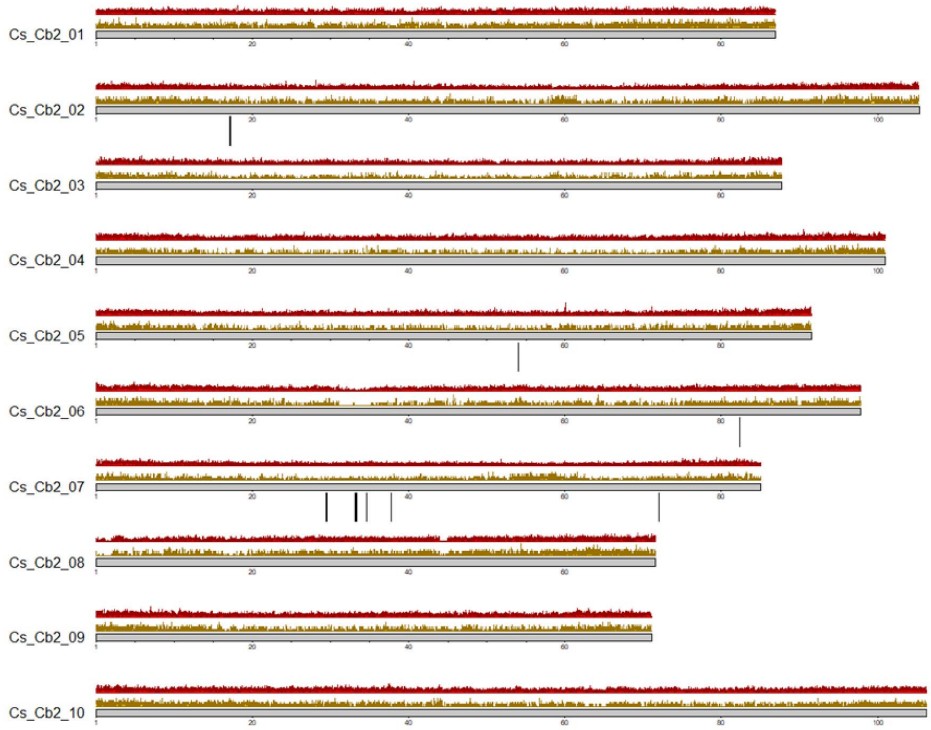

**Figure 6.** Cannbio-2 genome sequence assembly's karyoplot representing genome-wide density of masked repeat regions (gold), gene density (red) and regions of putative THC/CBD synthase genes (black lines).

## DATA AVAILABILITY

Sequence data has been deposited at DDBJ/EMBL/GenBank under the BioProject ID PRJNA667278. The Cannbio-2 sequence reads (short reads and long reads), genome assembly (draft genome assembly sequence and cs10 guided genome assembly sequence), contigs tilling path to chromosomes table, genome annotation and additional files have been deposited in the *GigaScience* GigaDB repository [29].

## DECLARATIONS
## LIST OF ABBREVIATIONS

BUSCO: Benchmarking Universal Single Copy Orthologs; Cb-2: Cannbio-2; CBC: cannabichromene; CBD: cannabidiol; CBG: cannabigerol; DoH: Department of Health; EVM: EVidenceModeler; HGAP4: Hierarchical Genome Assembly Process; ODC: Office of Drug Control; PAF: pairwise alignment format; PK: Purple Kush; SMRT: Single Molecule Real Time; $\Delta^9$-THC or THC: delta-9-tetrahydrocannabinol; THCV: delta-9-tetrahydrocannabivarin.

## COMPETING INTERESTS

The authors declare that they have no competing interests.

## FUNDING

This work was supported by funding from Agriculture Victoria and Agriculture Victoria Services.

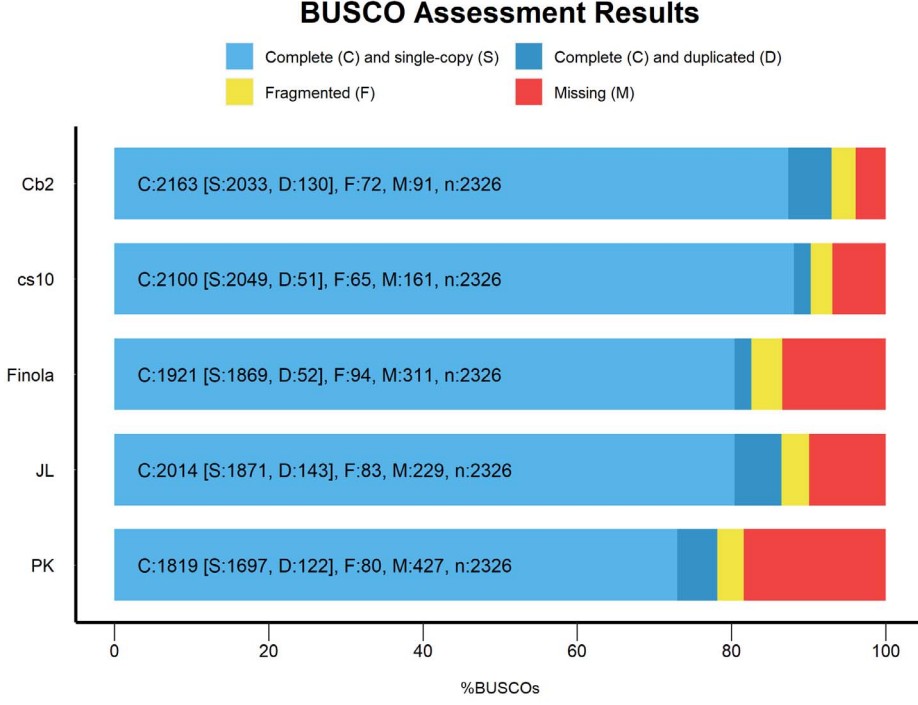

**Figure 7.** BUSCO evaluation results of Cannbio-2 genome sequence assembly from the current study as compared to published chromosome-scale whole genome sequence assemblies of cs10, Finola, JL and PK.

## AUTHOR'S CONTRIBUTIONS

S.B. and R.C.B. prepared plant materials, performed DNA extraction and sequencing of the libraries. S.B. conducted the data analysis and drafted the manuscript. N.O.I.C. assisted in the experimental design and data analysis. G.C.S. and N.O.I.C. conceptualized the project and assisted with preparation of the manuscript. All authors read and approved the final manuscript.

## ACKNOWLEDGEMENTS

The authors would like to thank Doris Ram, Alix L. Malthouse, Melinda C. Quinn and Larry S. Jewell for providing their support in the maintenance of the medicinal cannabis strains.

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
