## [Reviewer Report]

Comments on revised manuscriptI read through the revised article and they addressed some of the comments I made. My general comment, which they have not addressed satisfactorily (for me) is still the pseudochromosome assembly method.

From the start, only one method for reads de novo assembly was used. From our rice genome experience and current experience in tree genomes, we commonly use >1 to nominate a best method, or even combine outputs of the methods for improved scaffolding results. As you can see the N50 of their assembly is fairly small (187kb), it’s not convincing for a claim of high quality reference genome.

Anchoring the scaffolds by alignment to a known high quality reference for asserting scaffold ordering & pseudochromosome assembly, while this may be acceptable for animal genomes, is not enough for plant genomes. There’s a need to show that their particular cannabis accession (cb2) was most closely related to cs10, not to PK. This could easily be done with their reads and contig-level assembly data. By asserting scaffold order /anchoring to cs10 (which is a very different accession to cb2, as the authors pointed out in terms of chemotype), all subsequent cb2 genome comparison results is being ascertained with cs10 information. Without cb2-specific information (genetic map, optical mapping), the pseudochromosome assembly is not so solid.

Here’s a template paper they can refer to … https://www.nature.com/articles/s41597-020-0438-2 

For me, it would be more proper to report the genome as a draft assembly of 8,477 contigs, then include a contigs tiling path table vs cs10 or pk (depending on which accession cb2 is most closely related to). All the rest of the analyses on genome annotation would remain as-is, and is an important resource.

Additionally, reports on the repeat content (TEs, etc) to which they did the needed analysis already, genome heterozygosity /haplotigs would be very interesting findings to see, and the authors could easily do / may have done these analyses already, and just need to present these. Asserting low heterozygosity by casual observation is not enough, they have the data to measure this and report accordingly.

---

## [Reviewer Report]

Comments on revised manuscriptDifferent cs line could have variation in inversions, insertions, translocations, tandem repeats, using a different accession to anchor and order of contigs could introduced bias or even errors. Using linkage map or Hi-C data from the same accession to do scaffolding is better but needs a lot of lab and bioinformatics work, which may not be realistic.

The final genome assembly is significantly larger than the estimated genome size from a different cs line, I still suggest the authors do a k-mer analysis to validate the genome size further using Illumina or PacBio reads. The author of RaGOO said “RaGOO does not account for potential haplotype duplication in any way. Likely, such overlapping contigs will end up getting placed right next to each other in the final pseudomolecules. If one believes these duplications may exist in their assembly, I encourage them to use tools like purge_haplotigs prior to using RaGOO (https://github.com/malonge/RaGOO/issues/23).” Hence, the author needs to remove the potential redundancy in the final assembly.

---

## [Reviewer Report]

Upload additional filesDRR-20201001/form/17102020_SB_Cannabis_Reference_Genome_revised (1).docxReviewer name and names of any other individual's who aided in reviewer Wei ZhaoDo you understand and agree to our policy of having open and named reviews, and having your review included with the published papers. (If no, please inform the editor that you cannot review this manuscript.)YesIs the language of sufficient quality?YesPlease add additional comments on language quality to clarify if needed
Are all data available and do they match the descriptions in the paper? NoAdditional CommentsThe BioProject PRJNA667278 is not accessible.Are the data and metadata consistent with relevant minimum information or reporting standards? See GigaDB checklists for examples <a href="http://gigadb.org/site/guide" target="_blank">http://gigadb.org/site/guide</a>YesAdditional CommentsIs the data acquisition clear, complete and methodologically sound?YesAdditional CommentsIs there sufficient detail in the methods and data-processing steps to allow reproduction?YesAdditional CommentsIs there sufficient data validation and statistical analyses of data quality? NoAdditional CommentsThe size of the final genome assembly is significantly larger than the estimated size, which is indicative of redundancy. I would suggest removing the potential haplotype redundancy further. I would also suggest a k-mer analysis to validate the genome size.
For a chromosomal assembly, the ratio of properly paired reads is lower than expected. Is the validation suitable for this type of data?YesAdditional CommentsIs there sufficient information for others to reuse this dataset or integrate it with other data?YesAdditional CommentsAny Additional Overall Comments to the AuthorRecommendationMajor Revision

---

## [Reviewer Report]

Reviewer name and names of any other individual's who aided in reviewer Ramil MauleonDo you understand and agree to our policy of having open and named reviews, and having your review included with the published papers. (If no, please inform the editor that you cannot review this manuscript.)YesIs the language of sufficient quality?YesPlease add additional comments on language quality to clarify if needed
Are all data available and do they match the descriptions in the paper? NoAdditional CommentsBioproject PRJNA667278 in NCBI appears to be still embargoed, a reviewer link would be helpful.Are the data and metadata consistent with relevant minimum information or reporting standards? See GigaDB checklists for examples <a href="http://gigadb.org/site/guide" target="_blank">http://gigadb.org/site/guide</a>NoAdditional CommentsSample provenance / passport information is lacking for the Cannbio-2 material. Outright mention of the source of RNAseq +TSA info in the methods would be helpful.
Same comment as above for Genbank bioproject.Is the data acquisition clear, complete and methodologically sound?NoAdditional CommentsIt's mostly clear from the DNA extraction, pacbio sequencing and primary assembly. 
The anchoring of the assembled contigs into pseudochromosomes using another published genome lack detail and only broadly mention the software used (RaGOO). This is a very critical step that will distinguish if the Cannbio-2 assembly is an improvement vs the mentioned genome assemblies (esp. cs10, PK); it's a circular argument if the genome assembly is ascertained against existing assemblies from other cannabis accessions and declared improved. As noted by the authors, there are differences (rather than inconsistencies) between the compared published genomes, and these may be inherent in each genome; any analyses on an assembly based on these would cause ascertainment bias.Is there sufficient detail in the methods and data-processing steps to allow reproduction?NoAdditional CommentsThe previous comment regarding anchoring of contigs to an existing genome applies to this as well.
Regarding genome annotation, is there any basis for the choice of annotation method, i.e. annotator software (Augustus), the consensus builder (EVN), and PASA ? MAKER (MAKER-P) and BRAKER are available pipelines, both being reported as good for plants, and GeneMark is a prediction software suite that excels in plant genome annotation.
Re, evidences for annotation, it appears that transcript de novo assemblies were used, but the RNAseq data was not incorporated in the prediction step. No orthologous protein databases appear to have been used as hints for gene prediction. These are just observations/suggestions to further improve annotation quickly.
In general, the annotation steps would benefit from a bit more detail for reproducibility, but I would say the annotation if done at the contig level would be very solid.Is there sufficient data validation and statistical analyses of data quality? NoAdditional CommentsOn the assembly itself, since there was no mention of the method for anchoring contigs into chromosomes, there is no information on how scaffolds are spaced along the genome, is it padding by a fixed # Ns? Are all assembled contigs anchored or are there unanchored ones? Again on the point of anchoring and ordering of contigs, ideally evidence from the same sequenced material would be the best to use (an example - genetic linkage map with sequence-based markers). Plant genomes are notorious for rearrangements (inversions, insertions, translocations, tandem repeats etc) even within species, and this appears to be the weakest evidence in this paper (how the contigs were anchored into chromosomes).
Re gene annotation, you can conduct the BUSCO on the predicted genes and report those as well. Again, results will reflect the outcome of the annotation method used. For BUSCO in general, I'd be cautious in comparing results across published genomes and it would be more informative during an optimization of the assembly methodology or testing different assembly methods (checking whether you are improving the assembly of the same underlying dataset). On this same topic, are the unmapped contigs from other assemblies used? The same question with the assembly done by the authors apply. 
Is the validation suitable for this type of data?NoAdditional CommentsMostly yes for the primary genome assembly. The pseudochromosome assembly analysis data validation is not convincing. If done at the contig level, the genome annotation would be solid.
Is there sufficient information for others to reuse this dataset or integrate it with other data?NoAdditional CommentsRecapping, missing are the biomaterial information,information on pseudochromosome assembly, explicit mention of genbank IDs for transcript assembly and RNAseq data used in annotation (instead of being in the reference) would improve re-use and integration.
On the chromosome nomenclature, I don't understand why the author doesn't mention the ongoing nomenclature being used by the community as reported in the NCBI cs10 refseq release.Any Additional Overall Comments to the AuthorI believe reporting on results based on the main evidences generated by the authors (in this current work and the previous one on transcriptome) would make this a stronger data release, i.e. contig/scaffold assemblies, the annotation of that based on your own RNAseq data . 
On a related note, have you tried using your short-reads data during assembly? Could your assembly have been improved if you used the Illumina data during assembly itself (hybrid assembly, scaffolding)?
Cannabis genomes are known to be highly heterozygous, a report of this would be easy to conduct from your assembly vs your reads dataset especially the short-reads and would be an important finding.RecommendationMajor Revision